# A Case of Bovine Eosinophilic Myositis (BEM) Associated with Co-Infection by *Sarcocystis hominis* and *Toxoplasma gondii*

**DOI:** 10.3390/ani13020311

**Published:** 2023-01-16

**Authors:** Filippo Maria Dini, Monica Caffara, Joana G. P. Jacinto, Cinzia Benazzi, Arcangelo Gentile, Roberta Galuppi

**Affiliations:** Department of Veterinary Medical Sciences, University of Bologna, Ozzano dell’Emilia, 40064 Bologna, Italy

**Keywords:** *Sarcocystis hominis*, *Toxoplasma gondii*, BEM, cattle, meat-safety, Apicomplexa

## Abstract

**Simple Summary:**

In this study, a peculiar case of bovine eosinophilic myositis (BEM) observed in a beef cattle is described. BEM is a specific inflammatory myopathy, often associated with *Sarcocystis* spp., with multifocal gray-green lesions that can lead to considerable economic losses and public health issues. Through histological, molecular, and serological analyses, we confirmed the first detection of *T. gondii* DNA in a case of BEM, associated with the coinfection by *S. hominis*. Molecular results highlighted DNA of both pathogens within the lesion, in healthy muscle and or in the meat juice pellets, drawing attention to the possible role that a co-infection of *T. gondii* with *Sarcocystis* sp. may play in evoking BEM lesions.

**Abstract:**

Bovine eosinophilic myositis (BEM) is a specific inflammatory myopathy, often associated with *Sarcocystis* spp., with multifocal gray-green lesions leading to carcass condemnation with considerable economic losses. Here is described a peculiar case of BEM that occurred in an adult (16 month) cattle, born in France, bred, and slaughtered in Italy at the end of 2021. On inspection, muscles showed the typical multifocal gray-green lesions that were sampled for, cytological, histological, and molecular investigations, while meat juice was subjected to IFAT for *Toxoplasma* IgG. Genomic DNA was extracted from lesions, portions of healthy muscle and from meat juice pellet and analyzed by PCR targeting 18S rDNA, COI mtDNA and B1 genes, and sequenced. The cytology showed inflammatory cells mostly referable to eosinophils; at histology, protozoan cysts and severe granulomatous myositis were observed. A BEM lesion and meat juice pellet subjected to PCR showed, concurrently, sequences referable both to *S. hominis* and *T. gondii*. Meat juice IFAT resulted negative for *T. gondii* IgG. Our findings highlight the first detection of *T. gondii* DNA in association with *S. hominis* in a BEM case, suggesting a multiple parasite infection associated with this pathology, although the actual role of *T. gondii* infection in the pathophysiology of the diseases should be clarified.

## 1. Introduction

*Sarcocystis* is an Apicomplexan parasite infecting several hosts including humans. Among the more than 200 *Sarcocystis* species at least six are recognized as infecting bovine muscular tissue, namely, *S. hirsuta, S. bovifelis*, *S. bovini*, *S. cruzi*, *S. hominis* and *S. heydorni*. Felids serve as definitive hosts of the first three species, canids are definitive hosts of *S. cruzi*, while *S. hominis* and *S. heydorni* are zoonotic [1,2,3]. 

Over the past few years, several species of *Sarcocystis* associated with eosinophilic lesions in bovine muscles have been reported. Their identification has been based on histological observation of the cyst wall thickness, separating the species forming thick-walled (2–7 μm) sarcocysts and thin-walled (<1 μm) sarcocysts [4]. The most common *Sarcocystis* species in cattle belonging to the first group are *S. hominis*, *S. bovifelis*, *S. bovini* and *S. hirsuta* [4,5,6], while the second group includes *S. cruzi* and *S. heydorni* [7,8]. As some of the abovementioned species forming thick- (*S. bovifelis* and S. *bovini*) and thin- (*S. heydorni*) walled cysts have been recently described, misidentification may have occurred in the past with reference to etiology of bovine eosinophilic myositis [9].

Bovine eosinophilic myositis (BEM) is a specific inflammatory myopathy characterized by typical grey-green lesions in cattle muscles, visible during *post-mortem* inspection [4]. Although the etiology of the eosinophilic myositis remains uncertain [9], this condition in cattle is often associated with *Sarcocystis* sp. infection and can lead to carcass condemnation [10]. It has been hypothesized that the pathogenesis is linked with cysts’ degeneration together with the establishment of a hypersensitivity response towards the parasite [11]. Supporting this hypothesis is the finding of intralesional *Sarcocystis* species inside the eosinophilic granulomatous lesions [4,5,7] and the ability of *Sarcocystis* sp. antigens to induce an eosinophilic granulocyte-mediated immune response [12]. However, except for the latter study, BEM has never been reported during experimental infection with *Sarcocystis* [11]. The prevalence of BEM is very low worldwide, ranging from 0.002 up to 5% [5], while the prevalence of *Sarcocysts* sp. infection in cattle is extremely high, with values in Italy ranging from 67.8% up to 95% [1,13,14,15,16]. A possible explanation of this discrepancy could be that BEM may be associated with one or more specific *Sarcocystis* species [5], including those that are zoonotic. For this reason, the correct identification of the species involved in BEM is crucial in order to assess the risk for the consumer of eating raw or undercooked meat [13,17].

So far sequencing the 18S rDNA has been widely used for *Sarcocystis* species identification, even if several authors pointed out that misidentification may occur due to high conservative characters of this gene, as for example among *S. bovini*, *S. bovifelis* and *S. hominis* [6,18,19,20]. Therefore, cytochrome C oxidase subunit I mitochondrial (COI mtDNA) gene has recently been exploited as a useful genetic marker for the Sarcocystidae and has proved to be useful in resolving unclear species boundaries of closely related *Sarcocystis* spp. in different hosts [2,18,21,22]. In the wake of this evidence, molecular techniques have been recently developed to clarify the identification of *Sarcocystis* species infecting cattle, confirming the higher discriminatory power of COI mitochondrial gene for *Sarcocystis* species identification [23]. A novel species-specific multiplex PCR assay for the simultaneous identification of all the species of the genus *Sarcocystis* reported in cattle in Italy has been recently developed by Rubiola et al. [19]. Through this new molecular approach, during an investigation on the presence of *Sarcocystis* species in BEM cases, the presence of *S. bovifelis* and *S. hominis* has been observed and seems to be considerably higher in specimens isolated from BEM condemned carcasses than in samples isolated from randomly sampled slaughter cattle [1]; anyway, *Sarcocystis* species seem to be predominant in different geographical areas [9].

Furthermore, possible co-infections between *Sarcocystis* spp. and other Apicomplexa parasites, such as *Toxoplasma gondii* have been described in cattle [24,25,26,27]. *Toxoplasma gondii* is a widespread zoonotic protozoon that can also lead to important economic impacts in livestock, causing mainly reproductive failure in small ruminants [28]. In contrast to small ruminants, cattle appear to be largely resistant to *T. gondii* infections and rarely showed the presence of tissue cysts [29]. Reports on clinical toxoplasmosis in naturally infected cattle are rare and comprised only abortions in association with the isolation of *T. gondii* from the fetuses [30].

Our study presents a severe case of BEM in a Limousine bull imported from France and fattened and slaughtered in Italy by characterizing in detail the gross-pathological, cytological and histological features and by identifying the associated etiology by serologic and molecular analysis. 

## 2. Materials and Methods

### 2.1. Source Material

Portions of muscle (*gluteus, semimembranosus,* and *semitendinosus*) from a clinically healthy 16-month-old Limousine bull were conferred at the Department of Veterinary Medical Sciences of Bologna University because of suspected sarcosporidiosis. The bull was born in France and imported to a fattening unit in the province of Verona (Italy) at the age of 10 months to be fattened and then slaughtered. No signs of disease were observed during the entire period of fattening in Italy and no differences in body weight were noticed when compared to the cohort animals. The carcass was rejected at slaughterhouse inspection due to the presence of macroscopic nodular green lesions on most muscle masses.

### 2.2. Gross Pathology, Cytology and Histology Investigation 

A gross-pathologic examination of the skeletal muscles of the Limousine bull was carried out. Smears and impressions of different lesion were performed, fixed and stained with May-Grunwald Giemsa. Samples from both affected areas and healthy muscle were collected for histology. The samples were fixed in 10% neutral buffered formalin, trimmed, embedded in paraffin wax, sectioned at 3–4 μm, and stained with hematoxylin and eosin (H&E) for histological evaluation.

### 2.3. Meat Juice Extraction and Serology 

Portions of the skeletal muscle (approximately 1 kg) were frozen in a plastic bag at −20 °C immediately after sampling and thawed at +4 °C overnight for collecting meat juice. After defrosting, approximately 1.5 mL of meat juice from the bag was transferred into sterile tubes (Eppendorf, Hamburg, Germany). Meat juice tubes were then centrifuged at 2500 rpm for 15 min to remove coarse particles, and the supernatant were tested by Immunofluorescence Antibody test (IFAT) by commercial antigen (Mega Cor Diagnostik, Horbranz, Osterreich) consisting of tachyzoites cultured on Vero cells and, as a conjugate, rabbit anti-bovine IgG (Sigma Immunochemicals, St. Louis, MO, USA) bound to fluorescein isothiocyanate (FITC) and diluted 1/300. An initial dilution of 1:4 (cut off) was used for the meat juice. The pellet was stored at −20 °C for downstream analyses. 

### 2.4. Molecular Investigations 

Genomic DNA was purified from different parts of the muscle: four samples from macroscopic lesions (ML) and four from muscle macroscopically healthy (HM), of 25 mg each, and one meat juice pellet (MJ), approximately 200 μL, using Pure Link ^®^ Genomic DNA Mini kit (Invitrogen by Thermo Fisher), according to the manufacturer’s protocol. 

A first screening end-point PCR targeting 18S rDNA gene of Apicomplexa was performed on all of the samples with the primers COC-1 and COC-2, as described by Hornok et al. [31]. Briefly, a reaction volume of 25 μL, containing 12.5 μL 2× Dream Taq Hot Start Green PCR Master Mix (Thermo Scientific), 9.5 μL ddH2O, 0.25 μL (1 μM final concentration) of each primer, and 2.5 μL template DNA were used. For amplification, an initial denaturation step at 94 °C for 10 min was followed by 40 cycles of denaturation at 94 °C for 30 s, annealing at 54 °C for 30 s and an extension at 72 °C for 30 s. A final extension was performed at 72 °C for 10 min. 

Additional Multiplex PCR assay, described by Rubiola et al. [19] was performed to simultaneously identify all of the species of the genus *Sarcocystis* actually reported in cattle in Italy, targeting 18S rDNA and COI mtDNA. The multiplex-PCR contained 2.5 μL of template DNA, 0.25 μL (0.5 mM) of each primer, Sarco Rev, Sar F, Hirsuta, Cruzi, COI HB, COI H and COI B, 12.5 μL 2× Dream Taq Hot Start Green PCR Master Mix to a total volume of 25 μL. The PCR assay involved a denaturation step at 95 °C for 3 min, followed by 35 cycles at 95 °C for 60 s, 58 °C for 60 s and 72 °C for 30 s and a final extension of 72 °C for 3 min. 

Finally, a Nested PCR targeting the glycerol-3-phosphate dehydrogenase (B1 gene) of *Toxoplasma gondii* was performed as described by Jones et al. [32]. First round of amplification included a denaturation step at 96 °C for 2 min, followed by 40 cycles at 93 °C for 10 s, 57 °C for 10 s, and 72 °C for 30 s. The second round of amplification involved a denaturation step at 95 °C for 2 min, followed by 40 cycles at 93 °C for 10 s, 62.5 °C for 10 s, and 72 °C for 30 s. Amplifications were performed in a T-personal thermal cycler (Biometra, Goettingen, Germany). In all of the abovementioned PCRs, water was included as a negative control.

The PCR products were electrophoresed on a 1% (for the first two assays) and 2% (for B1 nested PCR), agarose gel stained with SYBR Safe DNA Gel Stain (Thermo Fisher Scientific, Carlsbad, CA, USA) in 0.5× TBE. For sequencing, the amplicons were excised and purified by Nucleo-Spin Gel and PCR Clean-up (Mackerey-Nagel, Düren, Germany) and sequenced with an ABI 3730 DNA analyzer (StarSEQ, Mainz, Germany). All of the primers used in this study are reported in Table 1. 

The trace files were assembled with Contig Express (VectorNTI Advance 11 software, Invitrogen, Carlsbad, CA, USA), and the consensus sequences were compared with published data by BLAST tools (https://blast.ncbi.nlm.nih.gov/Blast.cgi, accessed on 5 December 2022). Sequence alignments were carried out by BioEdit 7.2.5 [33], while p-distance and maximum-likelihood (ML) tree (K2+G substitution model for both genes and bootstrap of 1000 replicate) were calculated by MEGA 7 [34]. The sequences obtained in this study were deposited in GenBank under accession numbers OQ184854-56 (18SrDNA) and OQ190466-67 (COI mtDNA).

**Table 1 animals-13-00311-t001:** Forward and reverse primers used in the different PCR assays.

	Primers	Gene	Primer Sequences	Product Length	Reference
18S Apicomplexa	COC-1	18S	AAGTATAAGCTTTTATACGGCT	300 bp	[31]
	COC-2		CACTGCCACGGTAGTCCAATA		
*Sarcocystis* spp. Multiplex PCR	Sarco_Rev	18S	AACCCTAATTCCCCGTTA		[15]
	SarF		TGGCTAATACATGCGCAAATA	200–250 bp	[35]
	Hirsuta		CATTTCGGTGATTATTGG	108 bp	[15]
	Cruzi		ATCAGATGAAAATCTACTACATGG	300 bp	
	COI_HB	COI	AATGTGGTGCGGTATGAACT		[19]
	COI_H		GGCACCAACGAACATGGTA	420 bp	
	COI_B		TCAAAAACCTGCTTTGCTG	700 bp	
B1 *Toxoplasma* nested PCR	I Round for	B1	GGAACTGCATCCGTTCATGAG		[32]
	I Round rev		TCTTTAAAGCGTTCGTGGTC-		
	II Round for		TGCATAGGTTGCAGTCACTG	96bp	
	II Round rev		GGCGACCAATCTGCGAATACACC		

## 3. Results

### 3.1. Gross-Pathological Findings 

On gross pathology, skeletal muscles showed multifocal, firm, cohalescent green-yellowish round lesions with a diameter ranging from 0.1 to 1.5 cm. Some of the larger lesions (1.0–1.5 cm in diameter) had necrotic yellow-green content (Figure 1a), and the smaller (0.2–0.5 cm in diameter) were whitish in color and appeared solid or released only little yellowish-white material when squeezed (Figure 1b). Based on the gross pathology, a diagnosis of severe multifocal chronic myositis was formulated. 

### 3.2. Cytological and Histological Findings 

On cytology, inflammatory cells mostly referable to eosinophils were noticed. 

Histologically, the muscle revealed extensive multifocal areas of necrosis with mineralization surrounded by fibrosis and inflammatory cells, mostly lymphoid cells (Figure 2a,b). In some areas, single muscle fibers were noted to be atrophic or with initial necrosis (in a longitudinal section made visible by hypertrophy, loss of transverse striation, and initial fragmentation) immersed in connective tissue with macrophages, giant cells, and lymphoid cells. Occasionally, the foci presented degenerate parasitic cysts surrounded by variable numbers of inflammatory cells (among which many eosinophils), or, at a later state, macrophages and giant cells forming a granulomatous lesion (Figure 2c). Intact protozoan cysts (morphologically referable to *Sarcocystis* sp.) were detected in the unaffected muscle (Figure 2d).

Histologically, the retrieved findings were compatible with a severe granulomatous myositis. 

### 3.3. Serological and Molecular Results

The meat juice IFAT resulted negative for *T. gondii* specific IgG at a 1:4 dilution. 

Concerning the molecular analyses, all of the PCR assays successfully amplified all of the three matrices examined. In detail, in the PCR targeting the Apicomplexa 18S rDNA, all nine specimens were positive, with a band of ~ 300 bp. Sequences were obtained from six samples (4 ML, 1 HM, 1 MJ) and a BLAST search gave 99.6% identity with *S. hominis* in five (3 ML, 1 HM, 1 MJ), and 99.6–99.3% *Hammondia hammondi*/*T. gondii* only from ML.

The same samples tested with the multiplex PCR specific for cattle *Sarcocystis* showed a band of 420 bp of *S. hominis*. To confirm the results, the COI mtDNA of two samples (1 ML, 1 MJ) were sequenced. A BLAST search gave a 99.5% (ML) and a 99.7% (MJ) identity with *S. hominis*.

Finally, the nested PCR targeting B1 gene of *T. gondii* showed amplification of all the nine samples, with an amplicon of approximately of 96 bp. Unfortunately, only one specimen (MJ) gave a readable sequence showing 100% identity with *T. gondii* (Table 2).

The p-distance of the 18S rDNA *S. hominis* specimens (HM, ML and MJ) showed 0% genetic variability to the same species retrieved from GenBank and used for building the ML tree. The interspecific p-distance was 0.2–0.3% with *S. bovini* and *S. bovifelis*, respectively, reaching 1.2–1.3 and 1.4% with *S. hirsuta*, *S. heydorni* and *S. cruzi*, respectively. Concerning the more variable COI mtDNA the intraspecific variability among *S. hominis* was 0.1–0.2%, while the interspecific divergence was 0.9–1% with *S. bovini* and *S. bovifelis*, respectively, and 1.6–2.5 and 2.3 % with *S. hirsuta*, *S. heydorni* and *S. cruzi*, respectively. 

The ML tree of both molecular markers showed the same topology (Figure 3a,b); our specimens (both HM and MJ) are included in the *S. hominis* cluster and closely related to *S. bovini* and *S. bovifelis*. Moreover, the *T. gondii* sample formed a cluster with the other member of the family Sarcocystidae (*H. hammondi*/*T. gondii* and *N. caninum*). 

## 4. Discussion

This paper reports for the first time co-infection by *S. hominis* and *T. gondii* in a severe case of BEM, resulting in carcass condemnation at slaughter. *Sarcocystis* spp. infection is common in cattle, but the development of BEM only occurs in some cases and seems to be linked to the presence of some *Sarcocystis* spp. [4]. In Italy, it has been seen to be associated mainly with *S. bovifelis, S. hominis* and *S. cruzi* [1]. Occasionally, as shown in the present report, muscular involvement can be extremely severe, causing carcass discard and high economic losses. The histopathological lesions observed in this study were similar to those described by Wouda [4] in a case of BEM in a beef cow due to *S. hominis*.

The correct identification of the etiological agents involved in BEM are of primary importance as the cattle muscle can be infected by some important zoonotic parasites, i.e., *Sarcocystis* spp. and *T. gondii*. In this view, the identification at species level of the former genus should be considered of primary importance in order to discriminate the zoonotic from the non-zoonotic species. In fact, *S. hominis* and *S. heydorni* are meat-borne zoonotic parasites that can be transmitted by eating raw or undercooked meat, a very common practice in all the southern regions of Europe [13,17,18].

In our study, depending on the method used, both *S. hominis* and *T. gondii* were detected in all of the three matrices tested (ML, HM and MJ). In particular by the species-specific multiplex PCR for *Sarcocystis* spp. [19], the *S. hominis*-related amplicon was detected in all matrices, as well as with the nested PCR targeting B1 gene of *T. gondii* [32].

The less specific PCR targeting of the 18S rDNA of Apicomplexa [32], did not allow for properly identifying *T. gondii* (BLAST result *H. hammondi/T. gondii*), which was confirmed only by the B1 specific gene.

Despite the two Apicomplexa detected in all matrices examined, sequences of good quality were obtained mostly for *S. hominis*, probably due to the high presence of this species also in healthy tissues (observed in histological sections).

Lastly, in our study, the sediment of meat juice was used as a target for the DNA extraction and amplification, and to the best of our knowledge this is the first time that such a matrix has been used for this purpose. Interestingly, only from this matrix were we able to obtain a readable sequence of *T. gondii* B1 gene, probably because the staring amount of parasite was higher than in lesions and in healthy muscle.

Therefore, based on our results, meat juice sediment proved to be a promising matrix to be tested for the molecular diagnosis of cysts-forming protozoans infecting bovine muscle, independently of the presence of BEM lesions.

Co-infections in cattle between *T. gondii* and *Sarcocystis* spp. have been already described [24,25,26,27]. To the best of our knowledge, this is the first report of the presence of *T. gondii* DNA in a case of BEM. However, its role in evoking the disease is unclear, as the infection in cattle is usually asymptomatic [28] and the reported cases are mainly related to reproductive failure [30]. Nevertheless, ingestion of *T. gondii* tissue cysts from infected meat is a major route of infection for humans, with consumption of raw or undercooked meat from infected animals considered a significant public health risk [36]. There is evidence suggesting the important role of the beef as a source of human infection [37,38,39]. The reported seroprevalence for *T. gondii* infection in cattle varies between different countries, for example, 83.3% (*n* = 504) in southern Spain [40], 45.6% (*n* = 406) in Switzerland [41], and 10.2% in Italian beef cattle [42].

Contrastingly to what happens in other animal species, in cattle the seroprevalence does not give a valid indication of the risk for human infection by eating meat because cattle can eliminate their tissue cysts while remaining seropositive [43]. Possibly, only recently infected animals, which have not yet developed antibodies, have a parasite load high enough to be detectable by direct assays [44]. This is confirmed also in the presented study in which the animal was seronegative for IgG against *T. gondii* by IFAT on meat juice, while *T. gondii* DNA was observed in muscle.

The finding of *T. gondii* DNA within the BEM lesions draws attention to the possible role that a co-infection of this parasite with *Sarcocystis* sp. may play in evoking BEM lesions. In the human context, although toxoplasmosis is not a well-recognized cause of eosinophilia, the literature suggests that associated factors, such as coinfection with other parasites or drug hypersensitivity, may play a role in the development of eosinophilia with acquired toxoplasmosis [45,46]. However, this is only a possible pathogenetic analogy, as these two host species (bovine and human) have a totally different susceptibility to *T. gondii*, with very different consequences of infection.

## 5. Conclusions

In conclusion, we described in this paper the first detection of *T. gondii* DNA in a severe case of bovine eosinophilic myositis associated with a coinfection by *S. hominis*. Molecular results highlighted DNA of both pathogens within the lesion in healthy muscle and or in the meat juice pellets, laying the groundwork for a possible etio-pathogenetic correlation between co-infection and the development of BEM. However, the role of *T. gondii* in the pathogenesis of eosinophilic myositis is completely unclear, and further studies on co-infection of *T. gondii* and *Sarcocystis* sp. in BEM cases are necessaryto understand the possible role of *Toxoplasma* in this condition that impacts food safety and the economy of the livestock sector.

## Figures and Tables

**Figure 1 animals-13-00311-f001:**
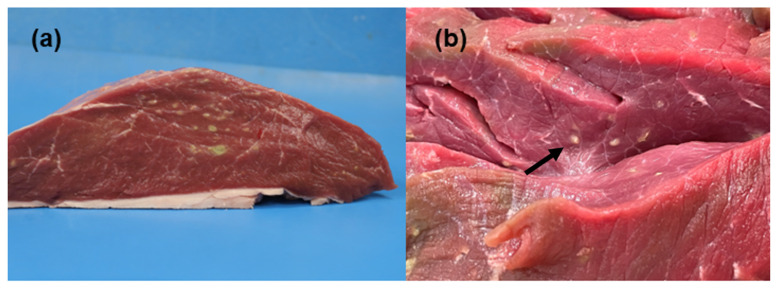
Macroscopic appearance of the infected skeletal muscle. (**a**) Note the multifocal yellowish-green round lesions, ranging from 0.1 to 1cm of diameter, with a necrotic content in a cross-section of skeletal muscle. (**b**) Note the white round lesions and a green discoloration area of the muscle (arrow).

**Figure 2 animals-13-00311-f002:**
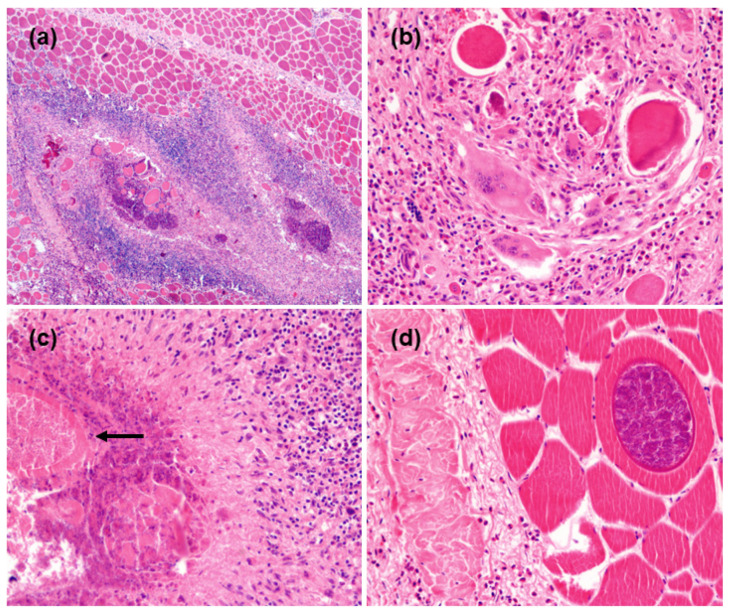
Histological appearance of the infected skeletal muscle. (**a**) A vast lesion showing mineralized foci of necrotic tissue surrounded by inflammation, mostly mononuclear elements. H.&E. 4× (**b**) Higher magnification shows a few necrotic fibers in the center of inflammatory tissue composed by palisades epithelioid macrophages, multinucleated giant cells, admixed with eosinophils and lymphoid tissue. H.&-E. 20× (**c**) On the left, extensive necrotic material in which the debris of a protozoan cyst are visible (arrow). Around them a layer of necrotic inflammatory cells surrounded by granulomatous tissue. H.&E. 10× (**d**) Top right: A cyst in the middle of a viable skeletal muscle fiber, with no inflammation. Left: Thick connective tissue (fibrosis) with scattered eosinophils. H.&E.10×.

**Figure 3 animals-13-00311-f003:**
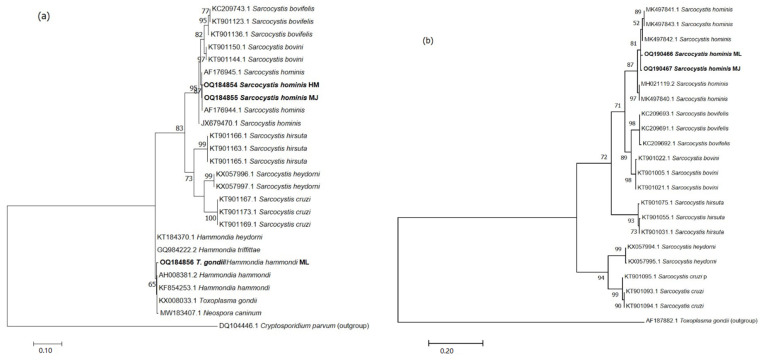
Maximum Likelihood trees inferred from Sarcocystis homins and Toxoplasma gondii 18S rDNA (**a**) and COI mtDNA (**b**) from this study and sequences retrieved from GenBank. The evolutionary history was computed with the Kimura 2-parameter model. The tree is drawn to scale, with branch lengths measured in the number of substitutions per site. There were a total of 207 position for 18S rDNA and 348 for COI mtDNA in the final data sets. Our specimens are in bold. HM = healthy muscle; ML = macroscopic lesion; MJ = meat juice pellet.

**Table 2 animals-13-00311-t002:** Results of PCR and Sequencing on tested samples.

Sample	18s PCR [31]	Multiplex Sarcocystis PCR [19]	B1 Toxoplasma Nested-PCR [32]
ML (4 samples)	4 PCR positive * (3 *S. hominis*, 1 *Toxoplasma/Hammondia*)	4 *S. hominis* PCR positive (1 *S. hominis*)	4 PCR Positive (no sequence)
HM (4 samples)	4 PCR Positive (1 *S. hominis*)	4 *S. hominis* PCR positive	4 PCR Positive (no sequence)
MJ (1 sample)	1 PCR Positive (1 *S. hominis*)	1 *S. hominis* PCR positive (1 *S. hominis*)	1 PCR Positive (1 *T. gondii*)

* In round brackets the results of Sanger sequencing.

## Data Availability

GenBank accession numbers OQ184854-56 (18SDNA), OQ190466-67 (COI mtDNA).

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
