# Peer review of "A Case of Bovine Eosinophilic Myositis (BEM) Associated with Co-Infection by Sarcocystis hominis and Toxoplasma gondii"

_animals, 2023, doi:10.3390/ani13020311_

Round 1

Reviewer 1 Report

The manuscript entitled “Molecular detection of Toxoplasma gondii and Sarcocystis hominis in a case of Bovine Eosinophilic Myositis (BEM): is coinfection related with BEM development?” by Filippo Maria Dini et al., describes a case report of bovine eosinophilic myositis in a Limousine bull whose carcass was condemned at slaughter due to the presence of macroscopic green lesions. Although reports of BEM are not new it Europe, nor in Italy, the in-depth investigation they performed, including a gross-pathologic examination of the skeletal muscles, a cytological and histological investigation, a meat juice extraction and serology and a molecular investigations including three different protocols (an end-point PCR targeting 18S rDNA gene of Apicomplexa, a Multiplex PCR assay targeting the 18S gene and the cox1 gene of all the Sarcocystis spp. described in Italy and a Nested PCR targeting the B1 gene of T. gondii) is undoubtedly appreciable. Besides, the detection of both S. homins and Toxoplasma gondii DNA is interesting and is worthy of attention and further investigation. Therefore, this manuscript deserves publication; however, a revision of the paper is required.

Required revisions are listed below.

Required revisions:

Title: I would remove the sentence “is coinfection related with BEM development?” as, although this case report is undoubtedly interesting, this is just a case report, and as you stated in your manuscript the role of Toxoplasma in the pathogenesis of BEM is completely unclear; furthermore, association is not correlation. The title “Molecular detection of Toxoplasma gondii and Sarcocystis hominis in a case of Bovine Eosinophilic Myositis (BEM)” it’s ok.

Line 13: Please remove comma as follows “BEM is a specific …”

Line 20: Please remove comma as follows “: Bovine eosinophilic myositis (BEM) is a specific inflammatory myopathy, …”

Line 35: Please replace “involved in” with “associated with”

Line 40: Please replace “included human” with “including humans”

Line 41: Please replace the symbol “>” with something like “Among the more than”

Line 47: S. bovifelis must been included as well among the species reported in association with BEM, as you mentioned in line 75 (Rubiola et al., 2021)

Line 66: Sarcocystidae must be italicized

Line 93: Here you could specify which muscles had BEM and were conferred to your Department (e.g. masseter, diaphragm etc)

Line 120: Here you could specify the amount of muscle you extracted (e.g 25 mg of four healthy muscle, 25 mg of four muscle affected with BEM etc)

Table 1: The table is unclear, can you better divide the different columns and lines (e.g. separating with a space or with a line the 18S Apicomplexa primers COC-1 and COC-2 from the Sarcocystis spp. Multiplex PCR primers and these primers from B1 Toxoplasma nested PCR primers).

Line 146: I would start a new paragraph as you performed the gel electrophoresis and the sanger sequencing on all your PCR products, not only on that obtained amplifying the B1 gene.

Line 194 (Serological and Molecular results): As you sequenced your PCR products for both S. Hominis and Toxoplasma gondii, did you deposit them in a public database, e.g. GenBank? Can you provide here the access numbers? It’s now required in any journal.

Figure 3: Just as a suggestion, I would consider removing your T. gondii sequences from the first phylogenetic tree and doing a third phylogenetic tree with your T. gondii sequences instead of including them in the 18S phylogenetic tree of S. hominis.

Lines 201-207: As sequencing of B1 gene failed in macroscopic lesion and healthy muscle, can you give an explanation or some hypothesis in the discussion session? Furthermore, as the macroscopic lesions resulted positive for the Apicomplexa 18S PCR, but the results “Hammondia hammondi/T. gondii”, you should stress the non-specificity of the fragment amplified by these primers. Finally, nor here neither in table two you mention how many samples were positive (e.g. all the four ML samples were positive for Toxoplasma/Hammondia and S. hominis or only some of them?)

Line 235: Remove “species” (as you already wrote “sp.”)

Line 236: Reformulate as follows: “Occasionally, as shown in the present report, the muscular …”

Line 252: Here you specify that T. gondii was found only in one lesion using the 18S PCR (one out of four ML samples), but this was not clearly specified in your results text (nor in table 2), can you specify it?

Lines 276-277: “The finding of T. gondii DNA within the BEM lesions draws attention to the possible role that a co-infection of this parasite with Sarcocystis sp. may play in evoking BEM lesions.” If I understand correctly you found and sequenced T. gondii only in meat juice, while the PCR targeting B1 gene gave positive results in PCR in all samples but sequencing failed, and the presence of Apicomplexa DNA referrable to Hammondia/Toxoplasma (PCR + sequences) was found in one single ML sample. In the discussion I would stress this part, as the molecular detection and investigation of Toxoplasma gondii was clearly not easy and forced you (correctly) to use different target genes which often didn’t give good results anyway.

Lines 284-288: I would reformulate this paragraph and I wouldn’t put it as the last paragraph of your discussion, as I think this is not the most important finding; indeed, your use of the sediment of meat juice is interesting, but it has some critical drawbacks, e.g. meat juice can’t give you the opportunity to selectively target the BEM lesions, which is quite important if you want to hypothesize the association of some microorganisms (Sarcocystis spp. or Toxoplasma) with BEM. I think this should be stressed in the overmentioned paragraph.

Line 296: Please replace “to confirm” with “to investigate”.

Author Response

Title: I would remove the sentence “is coinfection related with BEM development?” as, although this case report is undoubtedly interesting, this is just a case report, and as you stated in your manuscript the role of Toxoplasma in the pathogenesis of BEM is completely unclear; furthermore, association is not correlation. The title “Molecular detection of Toxoplasma gondii and Sarcocystis hominis in a case of Bovine Eosinophilic Myositis (BEM)” it’s ok.

The title has been modified.

Line 13: Please remove comma as follows “BEM is a specific …”

Done

Line 20: Please remove comma as follows “: Bovine eosinophilic myositis (BEM) is a specific inflammatory myopathy, …”

Done

Line 35: Please replace “involved in” with “associated with”

Done

Line 40: Please replace “included human” with “including humans”

Done

Line 41: Please replace the symbol “>” with something like “Among the more than”

Done

Line 47: S. bovifelis must been included as well among the species reported in association with BEM, as you mentioned in line 75 (Rubiola et al., 2021)

Done

Line 66: Sarcocystidae must be italicized

Even if the paper “Thines, M., Aoki, T., Crous, P.W. et al. Setting scientific names at all taxonomic ranks in italics facilitates their quick recognition in scientific papers. IMA Fungus 11, 25 (2020). https://doi.org/10.1186/s43008-020-00048-6” stated that all taxa should be written in italics, the “International Commission on Zoological Nomenclature” reported “The names of higher-ranking groups e.g. families or orders always begin with a capital but are not italicised”. https://www.iczn.org/outreach/guidelines-for-authors-and-editors/whats-in-a-name/

Line 93: Here you could specify which muscles had BEM and were conferred to your Department (e.g. masseter, diaphragm etc)

Done.

Line 120: Here you could specify the amount of muscle you extracted (e.g 25 mg of four healthy muscle, 25 mg of four muscle affected with BEM etc)

Done

Table 1: The table is unclear, can you better divide the different columns and lines (e.g. separating with a space or with a line the 18S Apicomplexa primers COC-1 and COC-2 from the Sarcocystis spp. Multiplex PCR primers and these primers from B1 Toxoplasma nested PCR primers).

Done

Line 146: I would start a new paragraph as you performed the gel electrophoresis and the sanger sequencing on all your PCR products, not only on that obtained amplifying the B1 gene.

Done

Line 194 (Serological and Molecular results): As you sequenced your PCR products for both S. Hominis and Toxoplasma gondii, did you deposit them in a public database, e.g. GenBank? Can you provide here the access numbers? It’s now required in any journal.

The sequences have been deposited to GB

Figure 3: Just as a suggestion, I would consider removing your T. gondii sequences from the first phylogenetic tree and doing a third phylogenetic tree with your T. gondii sequences instead of including them in the 18S phylogenetic tree of S. hominis.

Thanks for the suggestion, to avoid confusion we just change T. gondii with T. gondii/Hammondia hammondi

Lines 201-207: As sequencing of B1 gene failed in macroscopic lesion and healthy muscle, can you give an explanation or some hypothesis in the discussion session? Furthermore, as the macroscopic lesions resulted positive for the Apicomplexa 18S PCR, but the results “Hammondia hammondi/T. gondii”, you should stress the non-specificity of the fragment amplified by these primers. Finally, nor here neither in table two you mention how many samples were positive (e.g. all the four ML samples were positive for Toxoplasma/Hammondia and S. hominis or only some of them?)

The above comments have been addressed and the results have been modified as the table 2

Line 235: Remove “species” (as you already wrote “sp.”)

Done

Line 236: Reformulate as follows: “Occasionally, as shown in the present report, the muscular …”

Done

Line 252: Here you specify that T. gondii was found only in one lesion using the 18S PCR (one out of four ML samples), but this was not clearly specified in your results text (nor in table 2), can you specify it?

Done

Lines 276-277: “The finding of T. gondii DNA within the BEM lesions draws attention to the possible role that a co-infection of this parasite with Sarcocystis sp. may play in evoking BEM lesions.” If I understand correctly you found and sequenced T. gondii only in meat juice, while the PCR targeting B1 gene gave positive results in PCR in all samples but sequencing failed, and the presence of Apicomplexa DNA referrable to Hammondia/Toxoplasma (PCR + sequences) was found in one single ML sample. In the discussion I would stress this part, as the molecular detection and investigation of Toxoplasma gondii was clearly not easy and forced you (correctly) to use different target genes which often didn’t give good results anyway.

The above comments have been addressed

Lines 284-288: I would reformulate this paragraph and I wouldn’t put it as the last paragraph of your discussion, as I think this is not the most important finding; indeed, your use of the sediment of meat juice is interesting, but it has some critical drawbacks, e.g. meat juice can’t give you the opportunity to selectively target the BEM lesions, which is quite important if you want to hypothesize the association of some microorganisms (Sarcocystis spp. or Toxoplasma) with BEM. I think this should be stressed in the overmentioned paragraph.

Done, we moved the sentence above, specifying that this matrix could be useful not only in BEM cases.

Line 296: Please replace “to confirm” with “to investigate”.

Done

Reviewer 2 Report

L1 I suggest transferring type of publication to Case studies, since results are based on the investigation of BEM in single adult cattle

L1-406 Authors should read and cite review on Sarcocsytis in cattle and this parasite associated with eosinophilic myositis (https://doi.org/10.1016/j.ijpara.2022.09.009)

Some of important quotes  

Etiology of EM remains uncertain

Except in one report by Vangeel et al. (2012), discussed later, BEM has never been reported from cattle experimentally infected with Sarcocystis (Dubey et al., 2016a).

About Rubiola et al. 2021 study “In BEM condemned samples, the presence of S. bovifelis (46.3%) and S. hominis (40.7%) was significantly higher, while there was no statistically significant difference between the presence of S. cruzi (42.6%) or S. hirsuta (1.8%). The increased prevalence of S. bovifelis and S. hominis in samples condemned for BEM (of ~ 4 and 5-fold, respectively) adds to the suspicion of a possible causal link. However, no correlation can prove causation.

From the review of the BEM reports, it is evident that there is no clear association of a particular

species with BEM. In the USA, the thin- walled S. cruzi is the predominant species, both in the general

bovine population and in lesions of BEM. However, in Belgium, thick walled sarcocysts predominate. It

is also clear that the causal relationship could not be determined by a simple survey of Sarcocystis spp. in

cases of BEM and in normal populations.

L306-406 Latin genus and species names have to be written in italic

L20-36 from Animals Instructions for Authors “The abstract should be a total of about 200 words maximum.” Please shorten abstract. You can exclude “Bovine are common intermediate host of this protozoan, which also include zoonotic species such as S. hominis. It is hypothesized that the pathogenesis is linked with cysts degeneration, together with the establishment of hypersensitivity response towards the parasite.

L37 include S. hominis, it might be valuable for the visibility of the publication  

L42-45 complicated merge S. hominis with S. heydorni as zoonotic, S. cruzi has canids as definitive hosts and remaining three species are transmitted by felids. Also, mistake (L43) S. cruzi is transmitted via canids, and S. hirsuta via felids.

L46-47 explanation on what is thin-walled and thick-walled Sarcocsytis spp. should be provided. classification is based on histology. Thin-walled sarcocysts are S. cruzi and S. heydorni. Please be aware that all thin-walled sarcocysts prior to the description of S. heydorni in 2015 were attributed to S. cruzi. Also, prior to the description of S. bovifelis and S. bovini in 2016 all thich-walled sarcocysts were attributed to S. hominis and S. hirsuta. Taking into difficulties in species identification, the misidentification of Sarcocystis spp. in cattle appeared even after 2015-2016. In conclusion, association of S. hominis with BEM is overestimated.

L48, L77, L120, L124, L195-196, L200, L207, L208, L215, L219-223 not in MDPI style

L63 not precise citation. More et al. 2013; also add something after 2015 for instance (https://doi.org/10.1186/s13071-020-04473-9; https://doi.org/10.1016/j.ijpara.2019.05.008)

L68 add this citation (https://doi.org/10.1186/s13071-021-04788-1 )

L79 at the beginning of the sentence full species name should be given, Toxoplasma gondii

L81 it should be “ruminants [25].

L119 information on negative control in PCRs is missing

L153 the approximate length of amplified products would be beneficial

L191-192 transfer to L180-181

L196 if I understood correctly, all samples (4 ML + 4 HM + 1 MJ =9), so nine samples?

L196-218 must be re-written, improved.

L201 not clear, how many?

L202 the comparison of sequences with blast should be indicated in Methods section

L201-207

It is not clear how many specimens were positive using multiplex PCR and of these positive specimens how many gave good quality sequences, Table 2 can be improved showing positive number of examined isolates

L208-218 you should include in Methods how p-value was assessed, which software was used, same with ML phylogenetic trees. In the present study obtained sequences must be submitted to the GenBank.

L215 please improve wording “with our S. hominis specimens included in the S. hominis cluster and closely related with S. bovini and S. bovifelis”

L217 members of family Sarcocystidae would be more precise

L218 change to N. caninum

L219-223 indicate how many aligned nucleotide positions were in 18S rDNA and COI

L222-223 change to passive voice

L235-236 the prevalence of S. cruzi in BEM samples was also high. It was increased level of S. hominis and S. bovifelis prevalence in BEM samples comparing to the healthy ones, but there was no proves that BEM is associated with S. hominis and S. bovifelis. Therefore, I suggest here also to include S. cruzi.

L231-288 transferring type of publication to Case studies. Therefore, my suggestion would be also to shorten discussion, especially on the IgG against T. gondii.

L304 the obtained sequences must be available in GenBank

Author Response

L1 I suggest transferring type of publication to Case studies, since results are based on the investigation of BEM in single adult cattle

Even if the study is based on a single animal, we think that this research deserves to be published as full-length paper, considering the in-depth investigation performed, applying different diagnostic methods.

L1-406 Authors should read and cite review on Sarcocystis in cattle and this parasite associated with eosinophilic myositis (https://doi.org/10.1016/j.ijpara.2022.09.009)

The review has been cited

Some of important quotes  

“Etiology of EM remains uncertain”

“Except in one report by Vangeel et al. (2012), discussed later, BEM has never been reported from cattle experimentally infected with Sarcocystis (Dubey et al., 2016a).”

About Rubiola et al. 2021 study “In BEM condemned samples, the presence of S. bovifelis (46.3%) and S. hominis (40.7%) was significantly higher, while there was no statistically significant difference between the presence of S. cruzi (42.6%) or S. hirsuta (1.8%). The increased prevalence of S. bovifelis and S. hominis in samples condemned for BEM (of ~ 4 and 5-fold, respectively) adds to the suspicion of a possible causal link. However, no correlation can prove causation. ”

“From the review of the BEM reports, it is evident that there is no clear association of a particular species with BEM. In the USA, the thin- walled S. cruzi is the predominant species, both in the general bovine population and in lesions of BEM. However, in Belgium, thick walled sarcocysts predominate. It is also clear that the causal relationship could not be determined by a simple survey of Sarcocystis spp. In cases of BEM and in normal populations. ”

Thank for the suggestion. We added some sentences about this

L306-406 Latin genus and species names have to be written in italic

Done

L20-36 from Animals Instructions for Authors “The abstract should be a total of about 200 words maximum.” Please shorten abstract. You can exclude “Bovine are common intermediate host of this protozoan, which also include zoonotic species such as S. hominis. It is hypothesized that the pathogenesis is linked with cysts degeneration, together with the establishment of hypersensitivity response towards the parasite.”

Thanks for the suggestion. Done

L37 include S. hominis, it might be valuable for the visibility of the publication  

Done

L42-45 complicated merge S. hominis with S. heydorni as zoonotic, S. cruzi has canids as definitive hosts and remaining three species are transmitted by felids. Also, mistake (L43) S. cruzi is transmitted via canids, and S. hirsuta via felids.

The sentence has been changed.

L46-47 explanation on what is thin-walled and thick-walled Sarcocsytis spp. should be provided. classification is based on histology. Thin-walled sarcocysts are S. cruzi and S. heydorni. Please be aware that all thin-walled sarcocysts prior to the description of S. heydorni in 2015 were attributed to S. cruzi. Also, prior to the description of S. bovifelis and S. bovini in 2016 all thich-walled sarcocysts were attributed to S. hominis and S. hirsuta. Taking into difficulties in species identification, the misidentification of Sarcocystis spp. in cattle appeared even after 2015-2016. In conclusion, association of S. hominis with BEM is overestimated.

Thank for the suggestion, we have improved the sentence

L48, L77, L120, L124, L195-196, L200, L207, L208, L215, L219-223 not in MDPI style

Done

L63 not precise citation. More et al. 2013; also add something after 2015 for instance (https://doi.org/10.1186/s13071-020-04473-9 ; https://doi.org/10.1016/j.ijpara.2019.05.008 )

The sentence has been changed

L68 add this citation (https://doi.org/10.1186/s13071-021-04788-1 )

Citation added

L79 at the beginning of the sentence full species name should be given, Toxoplasma gondii

Done

L81 it should be “ruminants [25].”

Done

L119 information on negative control in PCRs is missing

Added

L153 the approximate length of amplified products would be beneficial

Done

L191-192 transfer to L180-181

Done

L196 if I understood correctly, all samples (4 ML + 4 HM + 1 MJ =9), so nine samples?

The exact number have been specified in the text

L196-218 must be re-written, improved.

All this part has been reorganized

L201 not clear, how many?

Specified

L202 the comparison of sequences with blast should be indicated in Methods section

Added

L201-207

It is not clear how many specimens were positive using multiplex PCR and of these positive specimens how many gave good quality sequences, Table 2 can be improved showing positive number of examined isolates

L208-218 you should include in Methods how p-value was assessed, which software was used, same with ML phylogenetic trees. In the present study obtained sequences must be submitted to the GenBank.

L215 please improve wording “with our S. hominis specimens included in the S. hominis cluster and closely related with S. bovini and S. bovifelis”

All the comments above have been addressed

L217 members of family Sarcocystidae would be more precise

Done

L218 change to Ncaninum

Done

L219-223 indicate how many aligned nucleotide positions were in 18S rDNA and COI

Done

L222-223 change to passive voice

Changed

L235-236 the prevalence of S. cruzi in BEM samples was also high. It was increased level of S. hominis and S. bovifelis prevalence in BEM samples comparing to the healthy ones, but there was no proves that BEM is associated with S. hominis and S. bovifelis. Therefore, I suggest here also to include S. cruzi.

Added

L231-288 transferring type of publication to Case studies. Therefore, my suggestion would be also to shorten discussion, especially on the IgG against T. gondii.

Please see comment above

L304 the obtained sequences must be available in GenBank

The sequences have been deposited in GB

Round 2

Reviewer 1 Report

Thank you for your revisions. The manuscript is now improved and deserves publication. I have just one last revision request, as a sentence is now unclear due to a revision:

Lines 45-50: Please reformulate as follow: “The most common Sarcocystis species in cattle belonging to the first group are S. hominis, S. bovifelis, S. bovini and S. hirsuta, while the second group includes S. cruzi and S. heydorni [7-8]. As some of the over mentioned species forming thick- (S. bovifelis and S. bovini) and thin- (S. heydorni) walled cysts have been recently described, misidentification may have occurred in the past with reference to aetiology of bovine eosinophilic myositis [9].

Author Response

Thank you for the suggestion, the sentence has been reformulated. 

Reviewer 2 Report

L40-41 I suggest to change to:

Felids serve as definitive hosts of first three species, canids are definitive hosts of S. cruzi, while S. hominis and S. heydorni are zoonotic.

L60-61 citation must be after the sentence

Author Response

Thank you for the corrections, changes have been added.